# Dolph2Vec: Self-Supervised Representations of Dolphin Vocalizations

## Abstract

Self-supervised learning (SSL) has opened new opportunities in bioacoustics by enabling scalable modeling of animal vocalizations without the need for expensive manual annotation. However, current SSL models in this domain prioritize broad generalization across species and are not optimized for uncovering the fine-grained structure of individual communication systems. In this work, we collect and release a novel dataset of over five years of longitudinal recordings, from five known dolphins in a semi-naturalistic marine environment—an unprecedented resource for studying dolphin communication. We adapt the Wav2Vec2.0 (1) architecture to this domain and introduce *Dolph2Vec*, the first large-scale, species-specific SSL model trained exclusively on this data. We benchmark our model on two biologically relevant tasks: signature whistle classification and whistle detection. *Dolph2Vec* significantly outperforms general-purpose baselines in both tasks. Beyond performance, we show that learned embeddings and codebook structure capture interpretable acoustic units aligned with dolphin whistle categories and possibly sub-whistle structure, enabling fine-grained analysis of communication patterns. Our findings demonstrate how SSL can serve as both a model and a scientific tool to explore hypotheses in animal communication research.

## 1 Introduction

Bioacoustics—the study of sound production, perception, and function in animals—is foundational for understanding animal behavior, ecology, and conservation (2; 3). A key application is the study of animal communication, which reveals social structures, cognitive abilities, and survival mechanisms (4; 5).

Among vocal species, dolphins are especially intriguing due to their sophisticated communication system. Their most studied vocalizations are tonal whistles (6), which include signature whistles (SWs)—individually distinctive sounds functioning as acoustic labels akin to names (7)—and non-signature whistles (NSWs), whose function remains unknown. Whistles are learned, mimicked (8; 9), and exchanged in sequences that maintain social bonds and coordination (10; 11). Despite these advances, our understanding of the function of dolphin whistles remains limited.

In recent years, deep learning (12) has become a pivotal tool in bioacoustics (13–16) by enabling scalable analysis of large audio datasets. Self-supervised learning (SSL) is particularly powerful for extracting structure from raw, unlabeled recordings (17–19). By designing proxy tasks that derive supervisory signals from the data itself, SSL removes the need for costly manual annotations—a major advantage in animal communication studies, where labeling is especially ambiguous due to the lack of ground truth.

Despite growing interest in SSL for animal vocalizations, most bioacoustic models remain general-purpose. Typically trained on large, heterogeneous datasets spanning a handful of vocalizations from many species—including diverse background and non-animal sounds (20–23)—they achieve broad generalization for tasks such as species detection and classification but dilute the species-specific structure needed to understand communication systems.

To address this limitation, we introduce the first large-scale, species-specific dataset of dolphin vocalizations: about 180,000 whistles collected over five years from a stable pod in a semi-naturalistic setting—up to three orders of magnitude larger than prior datasets. This longitudinal resource captures vocal behavior over time, enabling analysis of individual identity, social dynamics, and potential drift due to age or group reorganization.

Building on this dataset, we release *Dolph2Vec*, the first self-supervised model pre-trained exclusively on dolphin vocalizations. We adapt the Wav2Vec2.0 architecture (1) to dolphin whistle acoustics and benchmark it on a novel downstream task reflecting biologically relevant questions in dolphin communication. Unlike generalist models, our Wav2Vec-based architecture also enables hypothesis testing via learned codebooks, providing interpretable units grounded in the structure of the vocal signal.

More broadly, this work highlights the reciprocal value of combining deep learning with animal communication research. Animal vocalization datasets offer a rich testbed for developing and stress-testing machine learning models on species with acoustically rich, high-frequency, and continuously varying signals. Conversely, machine learning—particularly self-supervised models—provides a transformative approach to studying non-human communication, uncovering latent structure directly from raw data. These models can serve both as analytical tools and as hypothesis-generating engines in animal acoustic research. By demonstrating the power of SSL to reveal structure in bioacoustic data, we aim to strengthen the growing intersection of machine learning and animal communication and inspire new approaches to investigating the evolution and mechanisms of animal communication.

## 2  Related work

**Animal studies**   Dolphins produce three main sound types—echolocation clicks, burst pulses, and whistles—of which the latter two are central to communication (24; 25). A key element is the signature whistle (SW), an individually distinctive and stereotyped call used for identification and group cohesion (26). SWs, along with non-signature whistles (NSWs), constitute the majority of dolphin vocal output, with SWs accounting for up to 70% of whistles emitted in the wild (7). Recent work indicates SWs may include transient frequency modulations conveying information beyond identity (27), suggesting greater structural complexity than previously assumed, though their functional roles remain unclear.

Research on dolphin communication has largely relied on either behavioral studies of captive individuals in controlled environments (28; 29), or acoustic data from free-ranging dolphins (26; 30). The latter remains challenging to obtain, and, to the best of our knowledge, long-term, consistent datasets from the same individuals in the wild have not been released. Datasets typically provide short-term recordings of isolated instances of a mix of dolphin sounds (23; 31), or lack ecological realism due to captivity constraints. In addition, these datasets are often not publicly available (32). In contrast, our dataset consists of longitudinal recordings from a dolphin population living in a large, naturalistic marine environment we refer to as *semi-captive* (i.e. enclosed from boats but with openings to the sea). To our knowledge, this is the first publicly available dolphin dataset combining semi-captivity, longitudinal data, and whistle-level annotations, providing a new resource for studying both individual-specific and social aspects of dolphin acoustic communication.

**Deep learning for animal studies**   The growing availability of acoustic data (33) has enabled deep learning across bioacoustics (14–16), with spectrogram-based convolutional networks (34) widely used for detection, classification, and clustering. For dolphins, supervised whistle-classification approaches have been proposed (35; 36), but these rely entirely on labeled data and cannot uncover structure in an unsupervised fashion. While some studies have leveraged very large audio datasets to improve performance (37; 38), they still require vast amounts of annotated data, which is costly and labour-intensive to obtain.

Self-supervised learning (SSL) directly addresses this constraint by exploiting the abundance of unlabeled acoustic data. Instead of relying on human-provided labels, SSL models learn meaningful representations by defining proxy tasks that capture inherent audio patterns (see

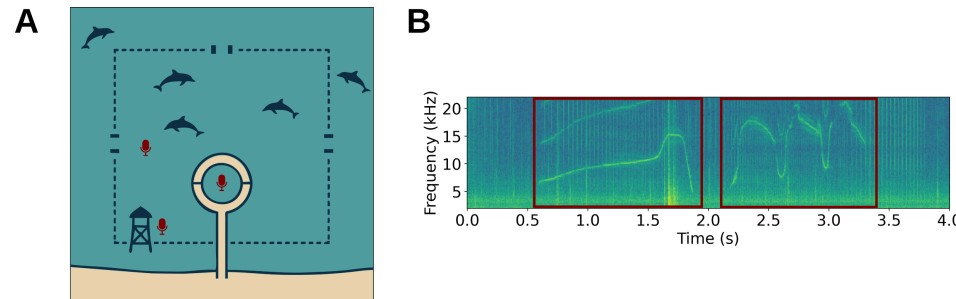

Figure 1: A) Schematic of the data collection setup, showing the dolphin area with three hydrophones and a rope with openings which allow dolphin passage to the open sea, while preventing boat access. B) A representative spectrogram of dolphin signature whistles showing distinct frequency patterns. The first example belongs to an individual named Nana, and the second one to Nikita.

Sec. A for details). This allows researchers to bypass annotation constraints and exploit large collections of raw recordings.

Models such as AVES (39), Nature-LM (22), and BioLingual (21) show that SSL can achieve strong downstream performance in species detection and classification. Nature-LM trains a generative audio–language model, while BioLingual uses a dual-tower audio–text approach with over a million synthetic captions—a powerful but less scalable strategy for homogeneous single-species datasets like ours. Our work instead uses an encoder-only architecture, well-suited for extracting representations for downstream tasks. Most similar to our work is AVES, which trains a HuBERT (18) model with several pre-training data mixes. While this model performs well for multi-species classification tasks, our focus is on an animal-specific model that shows good performance while also enabling testing theories grounded in biological studies of animal communication, an aspect overlooked in AVES (40; 41).

Transferability across domains remains unresolved. SSL models pretrained on human speech support species identification and call-type classification (42; 43), yet species-specific embeddings outperform general audio for birdsong (44; 45). WhaleLM (46) shows that SSL can also capture biologically relevant features in whale communication, while Gubnitsky et al. (47) stress species-specificity with a click detector for sperm whale codas. Our work contributes to this debate by introducing individual dolphin signature whistle identification with a dolphin-specific SSL model. *Dolph2Vec* combines large-scale pretraining with interpretable analyses to yield more generalizable and biologically informative embeddings.

## 3 Experimental Setup

In this section, we present the setup used for our data collection pipeline, the unique properties of our dataset, the pre-training of *Dolph2Vec*, as well as the data used for downstream tasks.

### 3.1 Data Collection

We present a novel dataset of bottlenose dolphin vocalizations collected in a semi-captive yet ecologically valid setting. Recordings were made in a natural marine enclosure in the Red Sea, where a resident pod of *Tursiops truncatus ponticus* coexists with human caregivers and visitors. Dolphins on the reef are untrained and free to enter and leave the area without restriction. This distinctive setup enables natural vocal behavior to be recorded while supporting long-term tracking of the same individuals (27; 48). Fig. 1**A** provides a schematic overview of the data-collection setup; a photo is shown in Appendix B.

The dataset consists of longitudinal acoustic recordings from four previously identified dolphins (48). In 2019, a fifth individual (*Tursiops aduncus*), an extralimital female from the Indian Ocean, joined the pod sporadically. Her signature whistle was identified from temporal

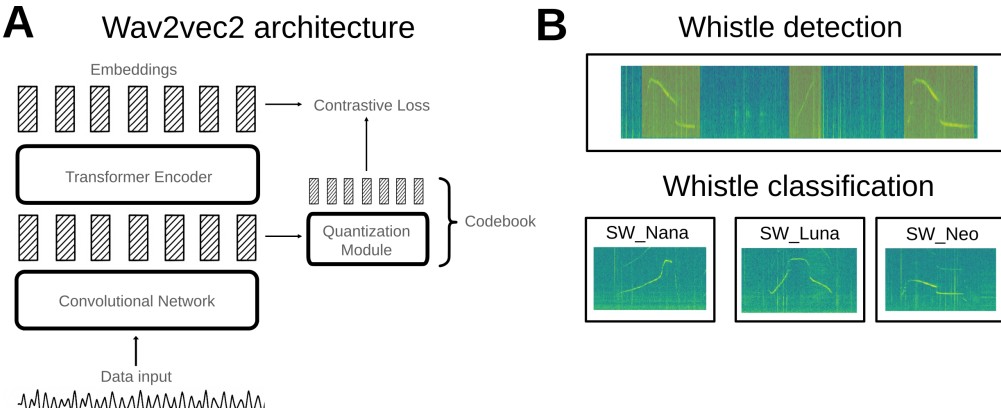

Figure 2: A) The Wav2Vec2.0 architecture used in *Dolph2Vec*. Raw audio is encoded into latent representations by a convolutional feature encoder, discretized via a quantization module with a learned codebook, and contextualized using a Transformer network. B) Downstream tasks: Top—whistle detection on spectrograms with highlighted whistles; Bottom—whistle classification of three distinct signature whistles from different individuals.

production patterns using the Signature Identification (SIGID) method (49). Additional information on the software and equipment used for data collection is provided in Appendix B.

Currently, the only large, publicly available dataset of dolphin sounds contains 566 dolphin whistles (23). Our dataset contains around 180,000 whistles, almost continuously recorded for 5 years, from the same and known pod of dolphins living in natural conditions. Furthermore, it includes whistle category labels for a subset of the data (around 8,000 whistles), enabling detailed analysis of vocal behavior over time. To support reproducibility, we will publicly release all data, providing a valuable resource for animal communication and computational bioacoustics research.

### 3.2 PRE-TRAINING DATA

Our pre-training dataset contains 100 hours of audio resampled at 44.1 kHz, spanning all recordings from October 2019 to April 2024 (excluding 2022 due to technical issues which prevented stable recordings). This corresponds to roughly 180,000 individual whistles, estimated from the labeled subset using empirical whistle durations and inter-whistle intervals (both 1 s). Under this assumption, 100.73 hours of recordings yield about 50 hours of whistling. The total amount of whistles is roughly 300 times more than other existing dolphin datasets.

To select pretraining data, we use a custom convolutional neural network (CNN) based on the VGG16 architecture (50) pre-trained on ImageNet (51) to extract recordings containing vocalizations. The CNN takes spectrograms as input and was fine-tuned on over 8,500 whistles identified by a custom algorithm leveraging spectral features and dynamic time warping (DTW) to known whistle templates (27). The resulting dataset comprises 33,267 segments (max duration 246 s), truncated to 20 s during training.

### 3.3 *DOLPH2VEC* - ARCHITECTURE

Unlike written language, which provides discrete symbols as natural supervision, audio is a continuous signal without predefined units. *Dolph2Vec* follows the Wav2Vec2.0 architecture (1), illustrated in Fig. 2A. It consists of a convolutional feature encoder, a quantization module, and a Transformer-based context network. The encoder processes raw audio into latent representations, which are discretized by the quantization module into learned codewords drawn from a codebook. These discrete units serve as targets in a contrastive SSL task, where a context network captures temporal dependencies to learn high-level speech features without labels. A diversity loss promotes balanced codebook usage.

### 3.4 DOLPH2VEC - TRAINING

We pre-trained a modified Wav2Vec2.0 model (1) on our dolphin vocalization dataset for 400K steps using 32 A100 GPUs with two steps of gradient accumulation and a batch size of 4 sound files per device (total $64 \times 4 = 256$). We used two codebooks with 320 vectors each and trained with the AdamW optimizer (52). Additional details on the pre-training setup and hyperparameters are in Sec. D.

Because our recordings are sampled at 44.1 kHz rather than the standard 16 kHz of speech, we modified the feature encoder to preserve the original model's temporal resolution. Specifically, we increased the first convolutional layer's kernel size from 10 to 30 and stride from 5 to 15, matching the receptive-field granularity of the original Wav2Vec2.0. All other architectural parameters follow the base configuration; see (1) for details.

### 3.5 DOWNSTREAM TASKS

We test the models on two downstream tasks of increasing granularity. First, we follow the detection task proposed in (39), and identify the presence or absence of specific whistle types in fixed-length audio segments. Secondly, we perform classification by assigning a label to isolated whistle sounds. Task representation is in Fig. 2**B**. For both tasks, we trained a logistic regression model with 5-fold stratified cross-validation using scikit-learn (53). We use the lbfgs solver and a maximum of 1500 iterations to ensure convergence. We use stratified splits to maintain the class distribution across folds. We test L2 regularization parameters with value 0.1, 1.0 and 10 and report the best score. Classification performance is measured by average accuracy across folds, and detection by mean average precision (mAP), following (20).

**Detection**  Detection is the task of predicting whether a whistle is present in 0.5-second audio segments, by also classifying its whistle category. Each segment receives binary labels for all known whistle types; segments without any are labeled non-whistle. The dataset was built by segmenting recordings and assigning labels based on annotations from the classification task, supplemented with manually labeled data from (23). The final dataset consists of 660 segments containing at least one labeled whistle, along with additional segments containing no whistles to serve as background.

**Classification**  For classification, we constructed a dataset of whistles labeled with the whistle type, containing 10 classes (5 signature whistles and 5 non-signature whistles). The classes were obtained by first classifying all whistles into categories using ARTwarp (27) (54), an unsupervised neural network algorithm which incorporates dynamic time warping (55). The automatically assigned labels were then manually corrected by expert annotators following visual inspection of spectrograms. Since the original dataset was highly imbalanced, with four classes having fewer than 300 samples each, we excluded these four categories. We then randomly sampled 500 instances from each of the remaining classes, creating a balanced dataset consisting of six classes. We use this dataset to train a linear regression model with stratified 5-fold cross validation.

### 3.6 BASELINE MODELS

**Acoustic Baselines**  As acoustic baselines, we evaluated traditional hand-crafted features including spectral features (spectral centroid, spectral bandwidth, spectral contrast, and spectral rolloff), Mel-frequency cepstral coefficients (MFCCs), and mean spectrogram representations.

**BioLingual**  As a baseline, we include BioLingual, a contrastive language-audio model based on the CLAP-LAION architecture (56) which was trained on AnimalSpeak (21), a large-scale dataset comprising over one million captioned bioacoustic recordings from 25,000 species. Using audio-text alignment, BioLingual enables zero-shot retrieval and classification across taxa. We evaluate its performance on dolphin vocalizations using its pre-trained audio encoder without additional tuning.

| Feature Type | Whistle Classification | Whistle Detection |
|---|---|---|
| Chance level | 16.7 | 8.3 |
| Spectral Features | $34.2 \pm 0.01$ | $44.7 \pm 4.44$ |
| MFCCs | $47.2 \pm 0.02$ | $53.3 \pm 3.72$ |
| Mean Spectrogram | $61.6 \pm 0.02$ | $65.5 \pm 3.74$ |
| AVES-core [39] | $74.0 \pm 0.01$ | $64.5 \pm 3.44$ |
| AVES-bio [39] | $76.3 \pm 0.01$ | $63.9 \pm 2.03$ |
| BioLingual [21] | $74.5 \pm 0.01$ | $67.6 \pm 4.33$ |
| *Dolph2Vec* (ours) | $\mathbf{82.0} \pm 0.01$ | $\mathbf{67.8} \pm 2.85$ |

Table 1: Accuracy on the whistle classification dataset and mAP on the whistle detection one. Scores computed using stratified 5-fold cross validation.

**AVES**  We also include AVES, a self-supervised transformer-based audio model adapted from HuBERT [57] and pre-trained on a large corpus of unannotated audio comprising animal vocalizations, human speech, and environmental sounds. AVES learns discrete latent targets through clustering and predicts masked waveform segments. We evaluate two variants: AVES-core, pre-trained on general audio datasets including FSD50K [58] and the balanced subset of AudioSet [37]; and AVES-bio, pre-trained on a curated subset of AudioSet and VGGSound [59] containing only animal vocalizations. We use the AVES encoder without further fine-tuning.

## 4   Results

### 4.1   *Dolph2Vec* - The First Large-Scale Species-Specific Self-Supervised Model

During training loss decreases steadily, with both contrastive and diversity losses contributing to this trend, as shown in Fig. 6. The declining contrastive loss indicates improved discrimination of latent representations, while the diversity loss ensures utilization of the full representational space. This confirms effective convergence of the pre-training process.

### 4.2   *Dolph2Vec* Matches State-of-the-Art Performance on Whistle Detection

Following standard practice in the field of self-supervised learning (60–62), after training *Dolph2Vec*, we froze its weights and use the model to extract embeddings for downstream tasks. Performance on these tasks acts as a measure of quality of model representations. Audio was resampled to 44.1 kHz models. Although the AVES models were originally pre-trained on 16 kHz inputs, we found that 44.1 kHz yielded better results on our data and was more appropriate given the frequency characteristics of our dolphin whistles dataset. For BioLingual, we retained the original 48 kHz sampling rate used during its pre-training to ensure compatibility and optimal performance.

Table 1 reports performance on the whistle classification and detection tasks across all feature types and embedding models. BioLingual and *Dolph2Vec* achieve the highest detection scores, with BioLingual at 67.6 mAP and *Dolph2Vec* slightly higher at 67.8 mAP, indicating that *Dolph2Vec* matches state-of-the-art performance on the whistle detection task.

### 4.3   *Dolph2Vec* Achieves New State-of-the-Art in Whistle Classification

Traditional acoustic features, used as baselines, achieved limited classification accuracy (Table 1, with spectral features reaching only 34.2% and MFCCs slightly higher at 47.2%. Mean spectrograms performed best among the baselines, achieving 61.6% accuracy.

Embedding-based models yielded significantly stronger results. AVES-core (39), AVES-bio (39), and BioLingual (21) all surpassed 70% classification accuracy. Our model, *Dolph2Vec*,

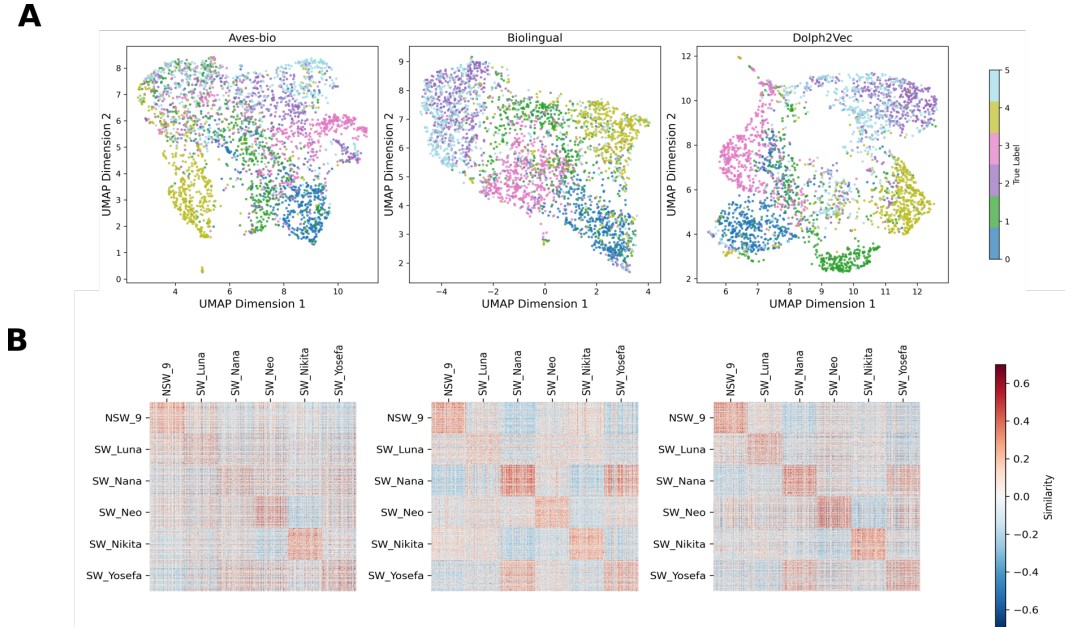

Figure 3: A) UMAP projection of learned embeddings from AVES-bio, BioLingual and *Dolph2Vec*, colored by their true label (Signature Whistle Category) B) RSA matrices of the three models.

achieved the highest overall performance, reaching 82% accuracy, demonstrating the strongest representation quality across both tasks. Notably, SW_Yosefa and SW_Nana are consistently confused by all models, reflecting their actual acoustic similarity and highlighting the biological realism of the benchmark. BioLingual was trained on the largest amount of data between all models while using an audio-text contrastive objective. While this showed remarkable performance on multi-species classification tasks (21), its inferior score on our task suggests that the model might not be able to pick up narrow features that are necessary when transferring to single-species benchmarks. This suggests a trade-off between broad and narrow performance when modeling bioacoustic data. Investigating this trade-off in relation to pre-training data is a valuable avenue for future work.

## 4.4 STRONG DISENTANGLEMENT OF SIGNATURE WHISTLE REPRESENTATIONS IN DOLPH2VEC EMBEDDINGS

To examine how well our model disentangles signature whistles from different individuals, we qualitatively and quantitatively analyze embeddings from AVES-bio, BioLingual, and *Dolph2Vec* using dimensionality-reduction techniques. We cluster UMAP projections of each model's representations with Gaussian Mixture Models (GMMs), which provide soft assignments and accommodate non-spherical cluster shapes, making them well suited to high-dimensional embeddings (Fig. 3).

*Dolph2Vec* embeddings show the clearest visual separation of the six ground-truth whistle categories (Fig. 3**A**). We further evaluate clustering performance with Adjusted Rand Index (ARI) and Normalized Mutual Information (NMI): *Dolph2Vec* achieves the highest scores (ARI = 0.3565, NMI = 0.4226), outperforming BioLingual (ARI = 0.2963, NMI = 0.3480) and AVES-bio (ARI = 0.1984, NMI = 0.2488). These results confirm that domain-specific self-supervised pretraining yields more structured and separable dolphin vocalization representations than general-purpose models.

To further characterize representational structure, we computed Representational Similarity Analysis (RSA) matrices between *Dolph2Vec* and the two baselines (63). RSA correlates pairwise similarity scores across models, capturing how their representational structures align.

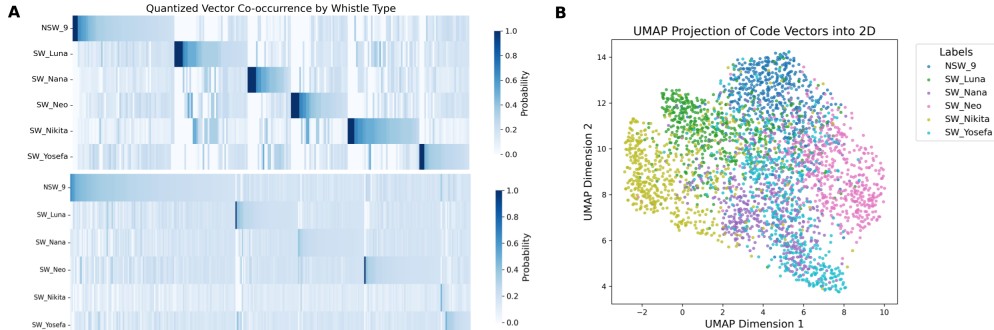

Figure 4: A) Codebook activations by signature whistle category in *Dolph2Vec* trained (top) and *Dolph2Vec* randomly initialized (bottom). B) 2D Projection of mean codevectors by SW category.

We computed RSA on unseen test data used for classification, reporting similarity matrices in Fig. 3**B**. *Dolph2Vec* shows stronger within-category similarity and clearer diagonal blocks (red) than baselines, indicating higher within-category consistency and better between-category differentiation. Cross-model Spearman correlations revealed that *Dolph2Vec* representations differed meaningfully from AVES-bio ($r_s = 0.35$, $p < 10^{-5}$) and BioLingual ($r_s = 0.31$, $p < 10^{-4}$), suggesting each model captures distinct statistical regularities.

## 4.5 Dolph2Vec Codebook Units Exhibit Partial Specialization for Signature Whistles

To test whether the discrete latent representations capture signature whistle (SW) information, we use *Dolph2Vec* to compute quantized latents $q_t$ and codebook indices $q_i$ across the full evaluation set, then calculate co-occurrence between annotated SW labels and codebook indices. Fig. 4**A** shows the conditional probability $P(\text{SW} \mid q_i)$: many discrete latents in *Dolph2Vec* (top) specialize for specific whistle types, unlike the randomly initialized model (bottom), which nonetheless retains some structure—consistent with prior findings on generalization in random networks (64). This may explain why specialization mainly occurs in the first codebook, while the second stays closer to its random state; all analyses in Fig. 4**A** therefore use one code set.

We then compute conditional entropy $H(\text{SW} \mid q_i)$ and mutual information $I(q_i; \text{SW})$, comparing against an untrained *Dolph2Vec* (Table 2) to quantify how strongly the learned latent space reflects class-specific encoding. Training *Dolph2Vec* markedly reduces conditional entropy and increases mutual information, indicating more informative and structured codebook representations.

The partial specialization of codebook indices—some highly specific, others shared across SW types—suggests they capture acoustic structure at a sub-whistle level. Fig. 4**B** supports this, showing that SW-type–averaged codevectors do not form distinct clusters, implying the learned codebook represents recurring acoustic features rather than whole whistle types. This aligns with hypotheses that meaningful information may occur at the sub-whistle level (27; 65; 8). We propose these learned units can act as fine-grained building blocks to investigate order effects and generate new hypotheses about dolphin acoustic communication

| Model | Conditional Entropy | Mutual Information |
|---|---|---|
| *Dolph2Vec*-Random | 2.13 | 0.43 (17%) |
| *Dolph2Vec* | 1.85 | 0.70 (28%) |

Table 2: Information-theoretic metrics comparing untrained and trained codebooks.

## 4.6 Perturbation of temporal features

To test how temporal structure contributes to individual-identity classification, we compared performance on original versus temporally shuffled vocalizations. Shuffling the feature-encoder output along the time axis preserves local acoustic content but disrupts global call sequence. Accuracy dropped from 82.0% on unshuffled input to 75.1% on shuffled input, indicating a modest but significant reliance on temporal structure. This suggests identity-relevant information is mainly encoded in short-timescale acoustic features, consistent with findings in human speech where models such as Audio-MAE (66) and WavLM (67) achieve above-chance performance even on temporally ablated inputs (40). The small effect further implies temporal structure is not pivotal for categorizing signature whistles. As future work, we propose systematically perturbing the frequency dimension (e.g., pitch shifting or spectral warping) to test its contribution, clarifying spectral versus temporal encoding strategies and informing hypotheses on key acoustic features in dolphin communication.

## 5 Limitations

While *Dolph2Vec* surpasses general-purpose models on a dolphin-specific task, its specialization compromises performance on multi-species or cross-ecological applications. Optimal performance on downstream tasks in a broad range of bioacoustic domains may be achieved by fine-tuning general models on large-scale, species-specific datasets, combining cross-species representational breadth with domain-specific granularity. The model focuses exclusively on acoustic features, omitting behavioral and environmental context critical for interpreting communicative function. Future integration of multimodal data—such as the individuals' movements, social dynamics, or environmental cues—will be necessary to ground acoustic signals in biologically meaningful events.

## 6 Conclusion

This work introduces the first large-scale dataset of dolphin vocalizations—over five years of longitudinal recordings from a pod of five dolphins in a naturalistic marine environment. With roughly 180,000 estimated whistles, it enables communication-focused research at a scale and resolution previously unavailable, bridging the gap between ecological realism and machine learning scalability.

We show that *Dolph2Vec*, a domain-adapted self-supervised model trained on this dataset, achieves state-of-the-art performance on new whistle classification and detection tasks. A large-scale, species-specific model can thus deliver both high performance and scientific insight. Analysis of *Dolph2Vec*'s internal structure reveals interpretable patterns in dolphin vocal behavior, including possible sub-whistle acoustic units—offering new ways to test hypotheses in animal communication.

Future work should explore domain-specific pretraining enhancements such as augmentations tailored to dolphin vocal features, adjusting the convolutional extractor and codebook to better match species-specific acoustics, and studying how human or background sounds in pretraining data affect performance. On the interpretability front, perturbing features such as frequency or duration could test classification robustness and clarify spectral vs. temporal encoding strategies. Another promising avenue is examining whether learned codebook units act as discrete building blocks in dolphin vocal sequences, shedding light on compositionality in dolphin communication.

Beyond technical advances, our findings highlight the mutual benefits of combining animal studies and deep learning. By releasing both our dataset and pre-trained model, we aim to catalyze cross-disciplinary research and promote integrative approaches to non-human communication, inspiring broader efforts to build species-specific resources and interpretable computational tools.

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

## A   ADDITONAL RELATED WORK: SELF-SUPERVISED LEARNING

The traditional supervised learning approach for dolphin vocalization embeddings has long been criticized for enforcing a human-biased perspective (68). This bias stems from linking each vocalization directly to expert annotations or predefined features assumed by humans to be important. For instance, analyses of dolphin whistles have often focused either on the

identity of the putative emitter or on the assumption that the whistle envelope is the primary carrier of information. However, this reliance on human interpretation and predefined notions risks overlooking crucial communication cues within the signal or misidentifying their significance, potentially missing the true underlying structure and meaning of the vocalizations.

To remedy this problem, unsupervised approaches may provide a new perspective that avoids human biases. Recent progress in natural language processing has demonstrated that the meaning and structure of language could be re-discovered through an unsupervised, machine-learning-based approach. Self-supervised approaches such as (69–72) have first proposed embeddings of words as vectors. These approaches are based on the distributional hypothesis (73): a word is defined by the context of its use. These unsupervised, token-based approaches are not directly applicable to domains where the unit of computation is less clear, like speech processing. Instead, speech-processing models like Wav2Vec2.0 (1) or HuBERT (18) simultaneously extract the unit of computation (speech units) and perform the contextual processing. The Wav2Vec2.0 architecture is composed of two processing blocks (Fig 2A): First convolutional layers extract the speech units through local computations. Next, a transformer block performs contextual processing. Learning is achieved by a masking objective, where the model should unmask speech units, with unmasking evaluated through a contrastive objective.

## B    DATA-COLLECTION SETUP

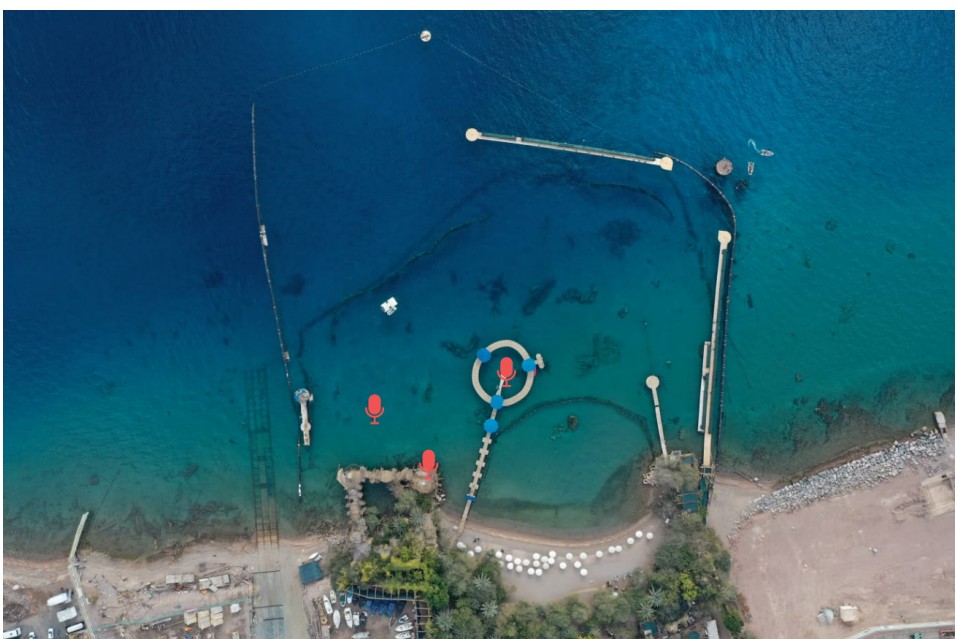

Figure 5: An aeral photo of our data collection setup.

The dataset was collected at Dolphin Reef in Eilat, Israel, a coastal site on the northern Gulf of Aqaba. This location serves as the natural habitat for a resident pod of bottlenose dolphins that freely move between the reef and open sea. Human-dolphin interactions occur only when initiated by the dolphins and are entirely voluntary. Hydrophones were placed in the locations as shown in Fig 5 Acoustic recordings were obtained using a set of 3 Brüel & Kjær® 8104 hydrophones connected to a 1704 preamplifier and a National Instrument® PCI-4474 acquisition card installed in a HP Z400 linux computer, sampling at 96 kHz, controlled by a custom-made code in C++. Recordings were conducted daily for one-hour periods at different times during the day (around 14 hours a day). The recordings were acquired between November 1, 2019, and March 12, 2024. Data acquisition was automated using scheduled crontab commands.

## C   Dataset properties

Studying a small, stable pod of five dolphins across several years provides advantages rarely available in animal communication research. The individuals' sex, family history, and kinship relations are well documented (27; 48), enabling integration of acoustic analysis with detailed social and historical context. The dataset thus combines individual-level identification with long-term recordings, supporting investigation of both fine-grained whistle structure and long-range social-linguistic patterns.

The pre-training dataset comprises 33,267 audio segments automatically extracted with a custom convolutional neural network. Segments average 12.91 seconds in length ($sd = 19.15$s), ranging from 2 to 246 seconds, and each contains at least one whistle. Whistles typically last about 1 second, with an average interval of 0.5 seconds between whistles, yielding an estimated total of roughly 180,000 individual whistles.

For the downstream classification task, a subset of about 8,000 whistles was annotated by domain experts through spectrogram inspection. These annotations distinguish signature whistles (SW) that serve as individual identifiers from non-signature whistles (NSW). The distribution of labeled data across categories is shown in Table 3.

| Category | Count |
| --- | --- |
| SW_Luna | 2,934 |
| SW_Neo | 2,239 |
| SW_Nikita | 888 |
| NSW_9 | 658 |
| SW_Yosefa | 626 |
| SW_Nana | 521 |
| SW_Dana | 335 |
| SW_Shy | 81 |
| NSW_3 | 45 |
| NSW_6 | 27 |

Table 3: Distribution of annotated whistle categories.

To balance categories, all classes with at least 500 examples were subsampled to 500 instances each (SW_Luna, SW_Neo, SW_Nikita, NSW_9, SW_Yosefa, SW_Nana), ensuring an even distribution for downstream evaluation.

## D   Pre-training setup

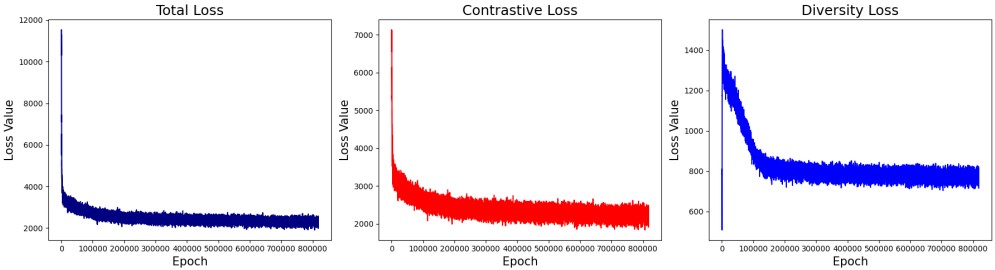

Figure 6: Pretraining losses over total training steps.

Training was conducted using the AdamW optimizer (52) with $\beta_1 = 0.9$, $\beta_2 = 0.98$, and $\epsilon = 10^{-6}$. The learning rate was set to $5 \times 10^{-4}$ with a linear decay scheduler and 32,000 warmup steps. Weight decay was fixed at 0.01. Training proceeded for a maximum of 400,000 steps with a per-device batch size of 4 on 32 GPUs. Mixed precision training was used to reduce memory consumption (74). The Gumbel quantization module (75; 76) employed a

temperature schedule starting at 2.0 and decaying multiplicatively by a factor of 0.999995 at each step, with a floor of 0.5. Fig. 6 shows the total loss, contrastive loss and diversity loss steadily decreasing, confirming convergence during training.

## E  ADDITIONAL BASELINES

| Feature Type | Whistle Classification |
|---|---|
| Chance level | 16.7 |
| Dolph2Vec2 (random init) | 37.9 |
| Wav2Vec2-base (pre-trained) | 47.0 |
| Dolph2Vec2-shuffled | 75.1 |

Table 4: Accuracy of additional baselines on our dolphin classification task.

We report performance on the signature whistle classification task for additional baselines in Table 4. Wav2Vec2 refers to the base model pretrained on human speech at 16 kHz (1). *Dolph2Vec*-random init denotes the *Dolph2Vec* model with randomly initialized weights, i.e., before any self-supervised pretraining. *Dolph2Vec*-shuffled is a variant of *Dolph2Vec* in which the temporal structure of the learned representations is disrupted by shuffling the output of the feature encoder along the time axis.

## F  BINARY WHISTLE DETECTION

| Feature Type | Detection (mAP) |
|---|---|
| AVES-bio | $99.93 \pm 0.13$ |
| AVES-core | $99.92 \pm 0.08$ |
| *Dolph2Vec* | $99.81 \pm 0.14$ |
| Mean Spectrogram | $99.82 \pm 0.10$ |
| BioLingual | $99.37 \pm 0.35$ |
| MFCCs | $98.17 \pm 0.54$ |
| Spectral Features | $95.94 \pm 2.69$ |

Table 5: Detection performance (mAP) on binary whistle vs. non-whistle task. Scores reported as mean $\pm$ standard deviation across stratified 5-fold cross-validation.

Table 5 reports performance on a binary whistle detection task, where the objective is to distinguish between whistle and non-whistle audio segments. Unlike the main detection task described in Section 3.5, which involves identifying specific whistle types in a multi-label setting, this task simplifies the problem to a single binary classification per segment. All models achieve near-ceiling performance, with mAP scores above 95, indicating that distinguishing whistle sounds from background noise is relatively easy. AVES-bio and AVES-core achieve the highest scores (99.93 and 99.92, respectively), followed closely by *Dolph2Vec* (99.81) and spectrogram-based features (99.82). BioLingual and other baseline features also perform well but slightly below the top models. Due to this performance saturation, the binary detection task provides limited insight into model differences and is included here for completeness.

## G  CODEBOOK SIZE ANALYSIS

We evaluated the hypothesis that a smaller codebook might better capture the structure of dolphin whistles by representing them as combinations of a limited set of sub-whistle units. To test this, we pre-trained several Wav2Vec models with varying codebook sizes: 32, 128, and 320 codewords per codebook. Model performance was then assessed across multiple downstream tasks and unsupervised metrics (codebook entropy, clustering quality).

The results indicate that the configuration reported in the main text, two codebooks with 320 codewords each, achieves the best balance of performance and representation quality. It outperforms smaller codebooks in downstream classification and detection tasks, while also producing superior entropy and clustering results. This suggests that a larger codebook provides a more accurate and flexible representation of dolphin whistle structure.

## H  SECOND CODEBOOK

Figure 7 shows that the second codebook is visually similar between the trained and randomly initialized models. Table 1 confirms this, with nearly identical conditional entropy and mutual information values. While the distribution of categories varies, no specialized or class-specific activation patterns emerge, indicating limited functional differentiation.

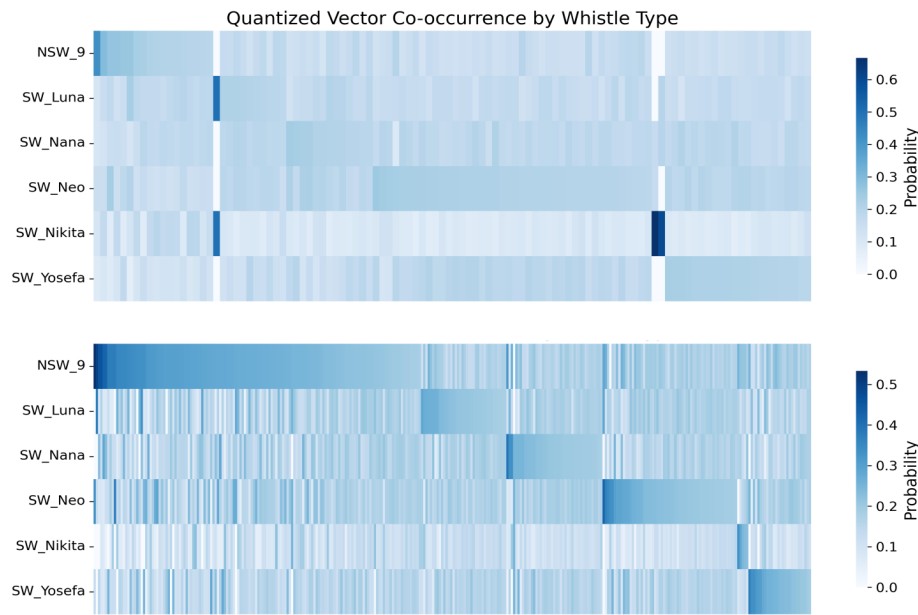

Figure 7:  Second Codebook activations by signature whistle category in Dolph2Vec trained (top) and Dolph2Vec randomly initialized (bottom).

|  | Conditional Entropy | Mutual Information |
| --- | --- | --- |
| *Dolph2Vec* | 2.5027 | 0.0687 |
| *Dolph2Vec* (random init) | 2.4675 | 0.0907 |

Table 6: Information-theoretic metrics for the second codebook.

