# OpenReview forum: "Dolph2Vec: Self-Supervised Representations of Dolphin Vocalizations"
_ICLR.cc/2026/Conference — Submitted to ICLR 2026_

### Official Review · Reviewer_WR6g · 2025-10-28

**Soundness:** 2
**Presentation:** 3
**Contribution:** 1
**Rating:** 2
**Confidence:** 5

**Summary:**

This paper introduces a new bioacoustic dataset of dolphin vocalizations, and a preliminary study applying a domain-specialized self-supervised learning (SSL) model, a pre-trained on Wav2Vec, to this new data. The work aims to advance the capabilities of audio models in the challenging domain of (marine) bioacoustics.

**Strengths:**

- _Novel dataset_: The primary strength of this work is the introduction of a new dataset in a challenging and interesting domain of marine bioacoustics. Bioacoustic data, particularly for complex vocalizations like those of dolphins, is difficult to acquire and to annotate, and this contribution could be a valuable resource for the community.
- _Clarity and readability_: The paper is well-written and easy to follow, making the authors' proposed contributions and methodology straightforward to understand.
- _Focus on an interesting problem_: The work addresses the hard task of modeling specialized bioacoustic audio, which is a significant and interesting direction for audio representation learning beyond common, general-purpose domains.

**Weaknesses:**

The paper, in its current form, has a number of weaknesses that limit its contribution.

**1. Literature review and positioning:**
- _Missing context and comparisons_: The related work section does not make a detailed comparison to other SSL approaches in bioacoustics, especially in avian bioacoustics (which seems to be the main driver in biacoustics/more mature) This is a critical omission, as numerous works directly address domain-specialized SSL and would serve as essential points of comparison (a quick search lead to e.g., Self-Supervised Learning for Few-Shot Bird Sound Classification [ICASSPW 2024] or Can Masked Autoencoders Also Listen to Birds? [TMLR2025]). Without this context, it is impossible to properly assess the novelty or significance of the paper's contribution.
- _Lack of dataset context:_ The paper does not provide a clear comparison of the new dataset to other existing bioacoustic datasets (e.g. Birb, BirdSet, BEANs etc.). A table detailing differences in scale, source, annotation, and acoustic characteristics would be good to properly compare the contribution (also for other marine datasets/collections etc.) .
- _Vague and missing citations:_ The paper makes unsubstantiated claims without specific references. For instance, the statement, "The total amount of whistles is roughly 300 times more than other existing dolphin datasets," is not supported by citations to the datasets being compared against (or further information in this context). In addition, the citation style appears inconsistent with the ICLR format.

**2. Baselines and experimental evaluation:**

- _Absence of baselines:_ Given that the paper's methodological contribution is very limited, a comprehensive and rigorous set of experiments is essential for the work to be impactful. The current evaluation is needs more necessary baselines. The authors should have compared their model against (e.g.):
    - Other established bioacoustic SSL and SL models from other domains (e.g., Perch, BirdNET, Bird-MAE, contrastive SSL models etc.). And marine models such as SurfPerch.
    - Other SOTA general-purpose audio models (e.g., supervised or self-supervised models pre-trained on large, diverse datasets like AudioSet (BEATs, EAT, etc.).
    - Include the W2V2 general-pretrained model in the main text (this is a very important comparison for the claims).

- _Limited evaluation protocol:_ The experimental protocol is minimal. A paper with a straightforward methodology requires a strong empirical evaluation to demonstrate its value. The current paper lacks standard protocols such as fine-tuning experiments. Furthermore, the model is not tested on any external datasets, which prevents any claims about its ability to generalize (as an interesting additional direction: how do the domain-specific models compete in other domains?).

**3. Methodological choices:**

- _Superficial method:_ The methodology consists of a basic application of a known model (W2V2) to new data. Given this lack of methodological innovation, a much more rigorous evaluation, including ablation studies and analysis, is expected to provide novel insights.
- _Missing motivation:_ The paper fails to provide a clear rationale for choosing Wav2Vec over other potential SSL architectures. There are no ablation studies to analyze the impact of different design choices, making the methodological decisions feel arbitrary.

**4. Dataset presentation and claims:**
- _Unclear data availability:_ For a paper whose primary contribution is a new dataset, it is a critical omission to lack a concrete plan for making it publicly available. The authors do not specify how, where, or when the data will be released. Also some dataset statistics in the main paper are missing to position the dataset properly.
- _Missing annotation details:_ The data collection and annotation processes are not detailed with sufficient clarity for others to understand the characteristics and potential limitations of the dataset.
- _Unsupported claims:_ The paper makes strong claims that are questionable in the context of the existing literature. For example, the claims that "current SSL models in this domain [bioacoustics] prioritize broad generalization" and that this is “The First Large-Scale Species-Specific Self-Supervised Model” appear to overlook the body of work on domain-specific avian models mentioned previously.

**Questions:**

- Given the existing body of work on domain-specific bioacoustic SSL, what is the specific novel contribution of this paper and what is the rationale behind the weaknesses mentioned above?
- Since the methodology is a direct application of W2V2, the paper's contribution rests heavily on its evaluation. Can you justify the choice of W2V2 over other SSL models and the omission of standard validation protocols like fine-tuning and ablation studies?
- As the dataset is a primary contribution, can you provide more detailed annotation procedures and key statistics? Crucially, what is your concrete plan and timeline for its public release to ensure the work is reproducible?

---

### Official Review · Reviewer_tBxu · 2025-10-30

**Soundness:** 3
**Presentation:** 3
**Contribution:** 3
**Rating:** 4
**Confidence:** 4

**Summary:**

The paper introduces Dolph2Vec, a self-supervised learning model based on
Wav2Vec 2.0, tailored for dolphins bioacoustic data. The authors introduce a new
dataset of dolphin vocalizations, collected over five years from five dolphins
in semi-natural conditions. Part of this dataset is annotated by experts to
facilitate evaluation on the task of whistle classification and detection. The
model is pre-trained on the unlabeled portion of the dataset and evaluated
using linear probing. The results demonstrate that Dolph2Vec outperforms
baseline bioacoustic models AVES, BioLingual and a Wav2Vec 2.0 model pre-trained
on general audio data (AudioSet). In addition, the authors analyzed the
embeddings and codebook activations.

**Strengths:**

- The paper releases a novel dataset of dolphin vocalizations, which is a valuable contribution to the bioacoustic community.
- The proposed Dolph2Vec model demonstrates strong performance compared to existing bioacoustic models.

**Weaknesses:**

- The novelty of the model is limited, as it mainly adapts the existing Wav2Vec 2.0 architecture to dolphin data without significant methodological innovations.
- The presented bioacoustic related work is lacking some recent references, especially regarding self-supervised learning in bioacoustics [1,2]. Also, it would be interesting to discuss relations to SurfPerch [3] and the transfer learning findings discussed in [4]. NatureLM-audio would be also an interesting baseline to compare against.
- The evaluation is limited to linear probing. It would be interesting to see how Dolph2Vec performs with fine-tuning or probing based on patch embeddings, this could improve performance further [1,2].

[1] Can Masked Autoencoders Also Listen to Birds? https://arxiv.org/abs/2504.12880
[2] Foundation Models for Bioacoustics -- a Comparative Review
https://www.arxiv.org/abs/2508.01277
[3] Leveraging tropical reef, bird and unrelated sounds for superior transfer learning in marine bioacoustics https://arxiv.org/abs/2404.16436
[4] Global birdsong embeddings enable superior transfer learning for bioacoustic classification https://arxiv.org/abs/2307.06292

**Questions:**

- As your dataset is a main contribution, have you considered giving it a distinct name to facilitate referencing in future work?
- Would you consider switching the description of "Detection" and "Classification" in 3.5 as you reference the latter in the first and to better align with your tables?
- How did you select the hyperparameters for pre-training and linear probing? Were they optimized on a validation set?
- How many random seeds were used for the experiments? What are the standard deviations of the results?
- Have you considered extending the pre-training dataset, e.g. with watkins marine mammal sounds?

---

### Official Review · Reviewer_ptAj · 2025-10-31

**Soundness:** 1
**Presentation:** 2
**Contribution:** 1
**Rating:** 0
**Confidence:** 4

**Summary:**

This paper introduces Dolph2Vec, a self-supervised system based on the Wav2Vec 2.0 architecture, that learns representations of audio containing dolphin vocalizations. A dataset of roughly 180k dolphin whistles from 5 dolphins was collected over a number of years and used to train Dolph2Vec. After training, two supervised classifications are run on the learned audio representations (one separately predicting each whistle category and another predicting from 6 specific whistle types). Baseline feature sets include a handful of spectral features, a pretrained contrastive learning model trained on bioacoustic data (BioLingual), and two pretrained variants of a self supervised audio model (AVES). Dolph2Vec is found to outperform the other models. UMAP projections of the learned embeddings and representational similarity analysis (RSA) visualizations show stronger clustering by true labels for Dolph2Vec than for AVES or BioLingual. Lastly, some of learned codebook vectors are found to specialize to the different whistle types.

**Strengths:**

1. Self supervised representation learning is a promising direction for bioacoustics and the study of animal vocalization.
2. Learning representations from raw audio carries potential improvements over some existing methods that only consider spectrogram representations.
3. The large, longitudinal dataset of dolphin vocalizations could aid the development of bioacoustics methods and the study of animal communication. I think this is a significant contribution.

**Weaknesses:**

1. The main weakness of the paper is its novelty. The proposed method is a straightforward application of Wav2Vec 2.0 to dolphin vocalization data, with little methodological innovation.
2. The quantitative comparison of the learned features on downstream classification tasks presented in Table 1 is lacking. Neither AVES nor BioLingual are trained or finetuned on the dolphin vocalization dataset Dolph2Vec is, so the superior performance of Dolph2Vec is uninformative. See Question 1.
3. The abstract states Dolph2Vec "can serve as ... a scientific tool to explore hypotheses in animal communication research." The codebook activation results presented in Figure 4A are the closest results in this direction, showing that some code vectors are selective to particular types of whistles. However, this is fairly unsurprising given the training data and does not seem to be motivate or explore any specific hypotheses about dolphin communication. Findings of this kind would greatly improve the paper.

**Questions:**

1. I am unsure what the intended use of Dolph2Vec is. Should we treat it as a general method that another researcher would use to study a large collection of vocalizations from a single species, or should it be treated as a dolphin-specific foundation model? If it is the former, it would be necessary to benchmark against alternative methods that are trained on the same dataset, including unsupervised methods. If it is the latter, it would be necessary to test the model's generalization to different individuals' vocalizations and varied recording conditions in a systematic way.
2. What are the dimensions of the input features used in the classification tasks? This information should be stated in the main text because it can have important consequences for linear separability.

Minor comments
* l125: Appendix A
* l201-206: this paragraph should call out an Appendix section with additional details, including how the 8500 whistles were selected.
* l220: I don't believe the second codebook is explained in the paper. I see one codebook in Figure 2A.
* l237: "report the best score" Isn't cross-validation used to determine the best L2 regularization parameter? This text makes it seem like all three regularizations were used.
* l256: do you mean "logistic regression"?
* l257: what are the hyperparameters being chosen in the cross validation scheme?
* Section 4.1: How were model hyperparameters chosen?
* l399: what statistical test is run? The text seems to suggest the null hypothesis is a correlation of 1.

---

### Official Review · Reviewer_FBZa · 2025-11-03

**Soundness:** 2
**Presentation:** 3
**Contribution:** 2
**Rating:** 2
**Confidence:** 5

**Summary:**

This paper introduces Dolph2Vec, a Wav2Vec model trained on bottlenose dolphin data. The authors also release a new, large-scale longitudinal dataset of approximately 180,000 dolphin whistles collected over five years from a known pod of five individuals. The authors report that Dolph2Vec achieves 82.0% accuracy, outperforming generalist baselines like AVES-bio and BioLinguagle , and suggest its learned representations capture interpretable, specialized acoustic units.

**Strengths:**

* Large-scale and valuable dataset of dolphin vocalizations
* In depth qualitative analysis of learned representations

**Weaknesses:**

I found this paper to have several substantive issues on two fronts. Firstly, data processing:

* The raw data is filtered to select dolphin whistles, per 3.2, "using a custom convolutional neural network" which in turn was trained on data that was selected "identified by a custom algorithm leveraging spectral features and dynamic time warping (DTW) to known whistle templates". This is a relatively complex data pipeline for which there is no validation. How do we know that the template matching and/or the CNN did not significantly bias the data? What is the retrieval/precision performance of this approach?
* The selected data is then classified, per 3.5, "by first classifying all whistles into categories using ARTwarp" and then "manually corrected by expert annotators following visual inspection of spectrograms". Again, what validation was performed on the ARTwarp method? What percentage of data had to be manually corrected? Did the annotators look at all the data or only spot-checked it? Were there multiple annotators? What is the annotator agreement?

Secondly, regarding the Dolp2Vec model evaluation:

* The literature shows that supervised bioacoustics models like Perch and BirdNET are very performant on non-avian data in general ([Ghani et al., 2023](https://arxiv.org/abs/2307.06292)) and on marine mammals in specific (see Watkins/BEANS perform in [Van Merriënboer et al., 2025](https://arxiv.org/pdf/2508.04665v1) and the results in [Burns et al., 2025](https://openreview.net/forum?id=xmZpX2ZWn2)). Generally speaking, BirdNET/Perch outperform AVES-bio and BioLingual for this setting, so I think it is important to compare to them in this work.

**Questions:**

* Can the authors please compare to BirdNET/Perch 2.0?
* Can the authors validate their template matching and custom CNN for whistle detection?
* Can the authors provide more details on the manual labelling process and validate it?

**Details Of Ethics Concerns:**

The data was collected at a private institution it seems (Eilat Dolphin Reef); does this affect the rights of the data?

---

### Meta-Review · Area_Chair_Me9x · 2026-01-06

**Summary:**

This paper proposes a dataset of dolphin vocalisations and trains a model via SSL to get efficient representations of this data. They eventually evaluate these representations in an identification task.

The authors have not engaged in rebuttal, and the reviews have placed this paper significantly below the acceptance threshold, so I recommend not accepting it.

**Reviewer Concerns:**

The authors have not engaged in rebuttal, and the reviews have placed this paper significantly below the acceptance threshold, so I recommend not accepting it.

**Reviewer Scores:**

The authors have not engaged in rebuttal, and the reviews have placed this paper significantly below the acceptance threshold, so I recommend not accepting it.

---

### Decision · Program_Chairs · 2026-01-26

Reject